# Do Apixaban Plasma Levels Relate to Bleeding? The Clinical Outcomes and Predictive Factors for Bleeding in Patients with Non-Valvular Atrial Fibrillation

**DOI:** 10.3390/biomedicines10082001

**Published:** 2022-08-18

**Authors:** Sutee Limcharoen, Manat Pongchaidecha, Piyarat Pimsi, Sarawuth Limprasert, Juthathip Suphanklang, Weerayuth Saelim, Wichai Santimaleeworagun, Pornwalai Boonmuang

**Affiliations:** 1Department of Pharmacy, Vajira Hospital, Bangkok 10300, Thailand; 2The College of Pharmacotherapy of Thailand, Nonthaburi 11000, Thailand; 3Department of Pharmaceutical Care, Silpakorn University, Nakhon Pathom 73000, Thailand; 4Division of Cardiology, Department of Medicine, Phramongkutklao Hospital, Bangkok 10400, Thailand

**Keywords:** non-valvular atrial fibrillation, non-vitamin-K antagonist oral anticoagulants, bleeding, thromboembolic

## Abstract

Apixaban can significantly prevent stroke events in patients with non-valvular atrial fibrillation (NVAF), as can be observed from the large, randomized, controlled trial conducted in the present study. However, the real-world evidence of bleeding events related to the apixaban plasma levels in Asian populations is limited. This study aimed to investigate the apixaban plasma levels and clinical outcomes among NVAF patients receiving apixaban, including determining the risk factors associated with bleeding during routine care. Seventy-one patients were included in the study. The median values were 112.79 (5–95th percentiles: 68.69–207.8) μg/L and 185.62 (5–95th percentiles: 124.06–384.34) μg/L for the apixaban trough (C_trough_) and apixaban peak plasma levels (C_peak_), respectively. Stroke and bleeding were found in 8 (11.27%) and 14 patients (19.72%), respectively. There was no statistical significance for C_trough_ and C_peak_ in the stroke and non-stroke groups, respectively. The median of C_trough_ (139.15 μg/L) in patients with bleeding was higher than that in the non-bleeding group (108.14 μg/L), but there was no statistical significance. However, multivariate analyses showed that bleeding history (odds ratio (OR): 17.62; 95% confidence interval (CI): 3.54–176.64; and *p*-value = 0.002) and C_trough_ (OR: 1.01; 95%: CI 1.00–1.03; and *p*-value = 0.038) were related to bleeding events. Almost all of the patients presented apixaban plasma levels within the expected range. Interestingly, bleeding events were associated with the troughs of the apixaban plasma levels and bleeding history.

## 1. Introduction

Apixaban is a non-vitamin-K antagonist oral anticoagulant (NOAC) used in stroke prevention for patients with non-valvular atrial fibrillation (NVAF) as a long-term form of treatment [1]. Current clinical practice guidelines strongly recommend the use of apixaban over warfarin, because large, randomized, controlled trials proved the net clinical benefit of these drugs in terms of their efficacy and safety [2,3]. Furthermore, apixaban has presented many advantages compared with warfarin, including rapid onset–offset, convenient dosing, predictable dose–response properties, few drug interactions, no requirement for international normalized ratio monitoring, and less frequent follow-up treatments [4].

Several studies have reported that apixaban is more effective in preventing stroke events or systemic embolism than warfarin in Asian and non-Asian patients with NVAF [5,6]. However, some studies determined that Asians have a higher exposure to apixaban than non-Asians, which could increase the bleeding risk in the former population because of genetic polymorphisms [7,8]. Apixaban is a P-glycoprotein (P-gp) substrate that is encoded by the ATP-binding cassette subfamily B member 1 (*ABCB1*) gene and presents inter-individual variability [9]. Previous studies have shown that *ABCB1* alters the apixaban plasma levels, especially C_peak_ [10], whereas apixaban activity can, at present, be measured as the anti-FXa activity and apixaban plasma level. Moreover, the therapeutic range of apixaban plasma levels is defined by the expected drug levels in clinical trials [7,11]. Subsequently, several studies have reported intra- and inter-individual variabilities in apixaban peak and trough concentrations, and that the apixaban concentrations were not associated with the clinical outcomes [12,13]. However, the real-world evidence of the related clinical outcomes of apixaban plasma levels in Caucasian and Asian populations is limited.

Consequently, this study aimed to investigate the clinical outcomes among NVAF patients in real-life clinical practice receiving apixaban, and the risk factors associated with bleeding events, especially the apixaban plasma levels.

## 2. Materials and Methods

### 2.1. Study Design and Patients

This study was a single-center, prospective cohort study conducted from January 2021 to January 2022 on patients who received apixaban in the Ambulatory Department of Phramongkutklao Hospital, a tertiary care hospital in Bangkok, Thailand. The study protocol was approved by the Institutional Review Board of the Royal Thai Army Medical Department and Phramongkutklao Hospital (Issued No.: Q021h/63).

Eligible patients were recruited with the following inclusion criteria: age ≥ 18 years with NVAF, receiving the appropriate dose of apixaban, and receiving apixaban continuously for at least seven days. All patients provided written informed consent before joining the study and were assessed for apixaban adherence at least seven days before enrollment via telephone and the pill-count method. The exclusion criteria were moderate-to-severe mitral valve stenosis, mechanical valve replacement, Child–Pugh class C of chronic liver disease, end-stage renal disease, pregnancy, breastfeeding, inability to care for themselves without a caregiver, poor adherence to medication, and receiving an inappropriate dose of apixaban.

### 2.2. Primary and Secondary Outcomes

The primary outcomes were the apixaban plasma levels and low-molecular-weight heparin (LMWH)-calibrated anti-Factor Xa (anti-FXa) activity at trough and peak concentrations. The expected ranges of the apixaban plasma levels were defined as 69–321 μg/L and 34–230 μg/L for C_peak_ and C_trough_, respectively [7,11]. The secondary outcomes were analyzed as follows: (1) stroke and bleeding events: stroke defined as ischemic and hemorrhagic strokes, including a definition of bleeding that was based on the criteria of the International Society on Thrombosis and Haemostasis (ISTH) (major bleeding was defined in nonsurgical patients as fatal bleeding, symptomatic bleeding in a critical area or organ, and/or bleeding causing a decline in the hemoglobin levels of 2 g/dL or more, or leading to the transfusion of two or more units of whole blood or red cells; minor bleeding was defined as “non-major bleeding”) [14]; (2) relationship between the clinical factors and bleeding outcomes; and (3) the correlation between the apixaban plasma levels and LMWH-calibrated anti-FXa activities.

### 2.3. Measurement of Apixaban Plasma Levels and Anti-FXa Activities

The steady-state apixaban plasma levels and anti-FXa activities were measured at their peak and trough concentrations by a chromogenic anti-FXa assay. Blood samples were collected at the peak and trough measurements. The current study defined the trough time as the time immediately preceding apixaban intake and the peak time as 2–4 h following the administration of the apixaban dose. Blood samples were collected into two 3.2% citrated tubes to determine both the trough and peak concentrations. The samples were centrifuged immediately for 15 min at 2500–3000× *g* [15,16]. Subsequently, the anti-FXa activity of the samples was measured using a chromogenic anti-FXa assay. Anti-FXa activity was determined using a Biophen^TM^ heparin LRT kit (Hyphen BioMed, Neuville-sur-Oise, France) and analyzed using a Sysmex CS 2500 System (Siemens Health Care, Milan, Italy). The anti-FXa activity results obtained from the chromogenic assay were converted into commercial apixaban (sensitivity range: 0–600 μg/L) and LMWH units (sensitivity range: 0–1.75 IU/mL) [16].

### 2.4. Statistical Analysis

Statistical analysis was performed using the Statistical Package of the Social Sciences Statistics version 27.0. (IBM Corp, Armonk, NY, USA). The baseline characteristics, apixaban plasma levels, and anti-FXa activities were analyzed using descriptive statistics. Categorical variables were presented as frequencies with percentages. Conversely, continuous variables were tested for normality following the Kolmogorov–Smirnov test and then reported as the means ± standard deviations (SDs) or medians with the 5th and 95th percentiles or interquartile ranges (IQRs). Student’s t-test or the Mann–Whitney U test were used to measure the continuous variables based on normally distributed continuous variables. More than two groups were analyzed using one-way ANOVA or the Kruskal–Wallis test.

The correlation between the apixaban plasma levels and clinical factors was analyzed using Spearman’s rank (ρ) and point biserial correlation (r_bp_).

The association between bleeding events and apixaban levels higher than the expected ranges was analyzed using chi-squared or Fisher’s exact tests. Consequently, logistic regression was used to predict any bleeding events.

The relationship between the apixaban plasma levels and LMWH-calibrated anti-FXa activities was analyzed using Spearman’s rank correlation (ρ) and simple linear regression. Statistical significance was considered at a *p*-value of <0.05.

## 3. Results

### 3.1. Baseline Characteristics of Patients

The baseline characteristics of all of the patients are presented in Table 1. Seventy-one patients were enrolled in this study. The mean age was 75.0 ± 10.5 years, and 36.6% of the patients were older than 80 years. Sixty-nine percent were male. More than half of the patients were >60 kg. The mean body weight was 67.3 ± 13.5 kg and the mean body mass index (BMI) was 24.61 ± 3.92 kg/m^2^. The median serum creatinine (SCr) was 1.1 mg/dL (IQR 0.9–1.4 mg/dL), and the mean creatinine clearance (CrCl) was 54.1 ± 22.0 mL/min. The majority of the patients had a moderate risk of stroke, the median value of the CHA_2_DS_2_-VASc score was 4 points (IQR 3–5), and the median value of the HAS-BLED score for bleeding risk was 2 points (IQR1–3). Forty percent had previously documented bleeding events. The most frequently reported sites of bleeding were the gastrointestinal tract (23.9%), cerebrovascular area (5.6%), urinary tract (5.6%), skin (4.3%), and oral cavity (2.8%). Hypertension (83.3%), dyslipidemia (70.4%), and anemia (42.3%) were the three most common comorbidities.

### 3.2. Steady-State Apixaban Plasma Levels and Anti-FXa Activities

The median times for the peak and trough plasma levels were 3.0 and 12.3 h, respectively. The median values (5th–95th percentiles) for the apixaban plasma levels were 112.79 (68.69–207.82) μg/L and 185.62 (124.06–384.34) μg/L at C_trough_ and C_peak_, respectively. Additionally, the trough (Xa_trough_) and peak (Xa_peak_) anti-FXa activities were 1.23 (0.59–2.21) IU/mL and 2.10 (1.44–3.11) IU/mL, respectively.

A total of 71 patients were divided into standard-dose (*n* = 46) and reduced-dose groups (*n* = 25). The apixaban plasma levels of the patients in the standard-dose group were 110.81 (63.08–249.01) μg/L and 205.44 (125.18–412.31) μg/L at C_trough_ and C_peak_, respectively. Xa_trough_ and Xa_peak_ were 1.23 (0.73–2.44) IU/mL and 2.21 (1.61–3.16) IU/mL, respectively. In contrast, no significant differences in the apixaban plasma level and anti-FXa activity at the peak and trough concentrations were observed between the standard- and reduced-dose groups. The steady-state apixaban plasma levels and anti-FXa activities are presented in Table 2.

We defined the expected range of the apixaban plasma concentrations as 34–230 μg/L at the trough and 69–321 μg/L at the peak levels, respectively [7,11]. Ninety-six percent of the patients had C_trough_ within the expected range, whereas 4% exceeded the expected range. The proportions of patients with C_peak_ within and above the expected range were 90% and 10%, respectively.

For the correlation between the apixaban plasma levels and clinical factors, only CrCl (correlation coefficient, ρ = −0.27; *p*-value < 0.024) and concomitant drugs with CYP3A4 or P-gp inhibitors (r_pb_ = 0.32; *p*-value < 0.006) were used as predictors of the apixaban trough plasma levels, whereas there was no correlation between the apixaban peak plasma levels and any other factors.

### 3.3. Stroke and Bleeding Events

Stroke and bleeding events were observed for one year. The trough and peak values of the apixaban plasma levels classified by stroke are also summarized in Table 3. Stroke events were observed in eight patients (11.27%), five patients were classified as having an ischemic stroke (7.04%), and three patients (4.22%) were classified as having a hemorrhagic stroke. The median value of C_trough_ was 146.24 (IQR 92.24, 163.22) μg/L in the stroke group and 111.48 (IQR 89.61, 156.92) μg/L in the non-stroke group, but the difference was not statistically significant (*p*-value = 0.604). Moreover, the median values of C_peak_ in the stroke group were 200.52 (IQR 161.61, 259.50) μg/L and 183.05 (IQR 146.00, 249.74) μg/L (*p*-value = 0.834). Overall bleeding events were observed in fourteen patients (19.7%) and classified by severity as major bleeding in six patients (8.5%) and minor bleeding in eight patients (11.3%). Most cases of major bleeding involved the gastrointestinal tract. The median values of C_trough_ were 139.15 (IQR 109.49, 163.16) μg/L in the bleeding group and 108.14 (IQR 87.23, 160.59) μg/L in the non-bleeding group, with no significant differences (*p*-value = 0.126). The median values of C_peak_ were 209.32 (IQR 145.23, 285.45) μg/L in the bleeding group and 183.05 (IQR 147.08, 247.03) μg/L in the non-bleeding group, with no significant differences (*p*-value = 0.470). The median values of C_trough_ in the minor, major, and non-bleeding groups were 149.46 (IQR 117.42, 172.89) μg/L, 123.95 (IQR 93.81–139.19) μg/L, and 108.14 (IQR 87.23–160.59) μg/L, respectively, with no significant differences (*p*-value = 0.148). The median values of C_peak_ in the minor, major, and non-bleeding groups were 269.30 (IQR 157.83, 317.69) μg/L, 178.66 (IQR 137.49–209.76) μg/L, and 183.05 (IQR 147.08–247.03) μg/L, respectively, with no significant differences (*p*-value = 0.330). The trough and peak values of the apixaban plasma levels classified by bleeding events are also presented in Table 4.

### 3.4. Relationship between Clinical Factors and Clinical Events

Based on the univariate analysis, there were eight factors that exerted potential effects on bleeding events, including hypertension, renal dysfunction, stroke, older age, drugs, alcohol, and the apixaban plasma level. The results of the univariate analysis suggest that the bleeding history is only one factor that is significant for increasing bleeding risk (OR 13.00; 95% CI 2.63–64.24; *p*-value = 0.002). As shown in Table 5, the multivariate analysis results demonstrated that the bleeding history (OR 17.62; 95% CI 3.54–176.64; *p*-value = 0.002) and C_trough_ (OR 1.01; 95% CI 1.00–1.03; *p*-value = 0.038) values are significantly related to bleeding events, while C_peak_ (OR 1.01; 95% CI 1.00, 1.02; *p*-value = 0.050) exhibited no significant difference in terms of bleeding events.

Additionally, in the univariate analysis, there were eight factors that exerted potential effects on stroke events, including hypertension, heart failure, previous stroke events, older age, vascular disease, diabetes mellitus, the CHA_2_DS_2_VASC score, and the apixaban plasma level. The results reveal that the CHA_2_DS_2_VASC score (OR 1.90; 95% CI 1.15–3.14; *p*-value = 0.013) is significantly related to stroke events. Unfortunately, the multivariate analysis results do not demonstrate any factors that exhibited significant differences in terms of stroke events.

### 3.5. Correlation between Apixaban Plasma Level and LMWH-Calibrated Anti-FXa Activity

The scatterplots (Figure 1) show a strong, positive linear relationship between the apixaban plasma levels and LMWH-calibrated anti-FXa activities, as confirmed by Spearman’s correlation coefficient. A statistically significant correlation was observed between C_trough_ and Xa_trough_ (correlation coefficient, ρ = 0.95; *p*-value < 0.001). Similarly, C_peak_ was significantly correlated with Xa_peak_ (ρ = 0.96, *p*-value < 0.001). In this study, a simple linear regression analysis was performed for the studied participants. The significant relationship between C_trough_ and Xa_trough_ (R^2^ = 0.86, *p*-value < 0.001) presented the following regression equation for the prediction of C_trough_:C_trough_ = 87.21 (Xa_trough_)

Likewise, C_peak_ and Xa_peak_ were significantly related (R^2^ = 0.90, *p*-value < 0.001), and the regression equation for C_peak_’s prediction of C_peak_ is as follows:C_peak_ = 129.48 (Xa_peak_) − 75.47

From the analysis, it can be observed that the slopes of the equation were different for C_trough_ and C_peak_. The scatterplots between the overall apixaban plasma levels (C_all_) and anti-FXa activities (Xa_all_) presented a curvilinear pattern. Therefore, the apixaban plasma levels were adjusted using a logarithmic scale. The simple linear regression was re-analyzed, and the equation was as follows:ln (C_all_) = (0.58) Xa_all_ + 3.99 (R^2^ = 0.91, *p*-value < 0.001)

## 4. Discussion

Our study aimed to investigate the apixaban plasma levels and clinical outcomes among Thai patients with NVAF who received apixaban as a form of medication. Approximately 90% of patients presented C_trough_ and C_peak_ within the expected range at the 5th–95th percentiles based on clinical practice guidelines [11]. The median values of the apixaban plasma levels were similar for C_trough_ and C_peak_, but slightly higher and lower when compared with the results obtained from Korean and Taiwanese patients [17,18]. In addition, the median values of C_trough_ and C_peak_ in the standard- and reduced-dose groups in the present study were lower and higher than the data obtained from the ARISTOTLE-J study, respectively [19]. Similarly, previous pharmacokinetic studies reported that apixaban exposure was 17.7% and 4.5% higher in Japanese and other Asian people, respectively [7]. However, the exposure to apixaban decreased by 3.2% in the Korean population [7].

The reduced-dose group showed a 10% higher median C_trough_ value and an 18% lower median C_peak_ value than the standard-dose group. However, the differences in the median values of C_peak_ and C_trough_ were not statistically significant in the standard- and reduced-dose groups. A previous study reported that the median values of C_trough_ and C_peak_ were approximately 25–30% lower in the reduced-dose group than in the standard-dose group [7]. Similar to the previous real-life studies, the results showed a 25–35% difference in C_trough_ and C_peak_ between the standard- and reduced-dose groups [9,12,20]. The results for the reduced-dose group presented a pharmacokinetic alteration, but the clinical outcomes were consistent in terms of efficacy and safety [21]. Interestingly, apixaban dose adjustment based on clinical practice guidelines among Thai patients with NVAF was maintainable within the expected range [11]. In this study, anti-FXa activity was slightly higher than that recorded in the data in a previous study conducted in Japan [22], since our study perhaps included some patients who concomitantly received CYP3A4 or P-gp inhibitors.

In addition, a previous study reported that apixaban concomitant with CYP3A or P-gp inhibitor drugs was not related to higher plasma levels in patients with AF [23]. However, our study presented a positive correlation between the concomitant CYP3A or P-gp inhibitor drugs and C_trough_. Amiodarone and dronedarone are moderate CYP3A4 and P-gp inhibitors that may increase the apixaban plasma levels [24,25]. These results conform to the results obtained from a previous pharmacokinetic study that suggested that strong to moderate CYP3A and P-gp inhibitors affect the AUC and apixaban plasma levels [26]. Furthermore, approximately 27% of apixaban was excreted unchanged by the kidneys. This pharmacokinetic finding illustrates that the impact of renal impairment on apixaban exposure is modest [27]. A regression analysis showed that AUC0–∞ was inversely related to apixaban clearance in renal impairment compared with healthy volunteers with normal renal functions [7,28]. Testa et al. provided data from a real-world study that determined that CrCl was poorly correlated with the apixaban plasma levels [9]. Similar to the present study, they observed a weak inverse correlation between CrCl and the trough apixaban plasma levels.

As the results of our study show, almost every patient had peak and trough apixaban plasma levels in the expected therapeutic range. It is noteworthy that the apixaban plasma levels in the stroke group had a higher trend than the non-stroke group, but no apixaban plasma levels were below the expected therapeutic range. Unlike in previous studies, the results revealed that the apixaban plasma levels were below the expected trough concentrations in 25–27% of patients affected by stroke events [29,30]. One potential explanation for why the apixaban plasma levels in the stroke group were higher than those in the non-stroke group may be related to low CrCl concomitant with amiodarone. Interestingly, all stroke cases in our study had previously suffered from stroke events in six to eight octogenarian patients. A history of stroke events and age ≥ 75 years were major risk factors for stroke events in AF patients [31]. As mentioned above, it may be that the apixaban plasma levels cannot be used to predict stroke outcomes, as suggested by the previous report [12]. In addition, a cohort study previously reported that thrombotic complications occurred in AF patients who had depleted apixaban plasma levels (22–145 µ/L) and high CHA_2_DS_2_VASc scores [32]. Therefore, the apixaban plasma level is the not only factor that predicts stroke outcomes in patients who receive apixaban—other factors, such as a history of stroke, age, hypertension, and diabetes mellitus, should be considered [33].

In the ARISTOTLE trial, a population pharmacokinetic model provided information for predicted apixaban levels in patients with or without bleeding events [34]. The analysis revealed a substantial overlap in the apixaban plasma levels between bleeding severity. The median plasma levels in each group only differed by approximately 15%, which caused difficulty in identifying a therapeutic range [34]. Subsequently, the results of our study are similar to those obtained in a previous study [35]. Explicitly, the bleeding group showed no difference from the non-bleeding group in terms of the median values of C_trough_ and C_peak_. Furthermore, the bleeding events were not associated with the expected therapeutic range listed above. However, the real-world pilot prospective study reported significantly higher C_trough_ and C_peak_ values in the bleeding group compared with the non-bleeding group [36], which is similar to our study. Concomitant with CYP3A4 and P-gp inhibitors, older age was a factor causing high apixaban plasma levels in the bleeding group [37,38]. Many reports demonstrated drug interactions and the effect of age on the apixaban levels [7,26,39].

In addition, we analyzed the model to predict the effects of bleeding history and C_trough_ on bleeding events. Similarly, the AVERROES study reported that C_trough_ was related to minor bleeding [40]; however, other previous clinical trials showed that the trough NOAC plasma level could predict the risk of bleeding [41,42]. Our study shared some differences from a previous observational study, which was the relationship between the bleeding events and C_peak_ [43]. Meanwhile, the present study observed no relationship between C_peak_ and bleeding events. At present, studies conducted on the correlation between the apixaban plasma levels and clinical outcomes are limited, since a real-world multicenter study confirmed high NOAC inter-individual variability.

In this study, we used chromogenic anti-FXa assays, and the Biophen^TM^ heparin LRT can be utilized as a surrogate reference standard [16,44]. In addition, the chromogenic anti-FXa assay is the most appropriate method to measure the pharmacodynamics of apixaban because of the strong linear correlation with the apixaban plasma levels [45,46]. Our study consistently observed a significant correlation between the apixaban plasma level and LMWH-calibrated anti-FXa activity. The real-life model in this study predicted that the linear relationships between the apixaban plasma levels and anti-FXa activities were R^2^ = 0.86 and R^2^ = 0.90 at the trough and peak, respectively. However, this relationship presented a curvilinear scatter plot between the C_all_ and Xa_all_, which was similar to the previous studies conducted using chromogenic assays and mass spectrometry [47,48]. All the patients presented anti-FXa activity within the range presented previously [27,49]. However, some patients in this study had apixaban plasma levels above the expected ranges. Thus, the LMWH-calibrated anti-FXa activity interpreted as the action of apixaban may be underestimated for the apixaban level.

It is important to note the limitations of this study. First, the sample size was insufficient to detect the relationship between the apixaban plasma levels and clinical outcomes. Second, minor bleeding events may not have been documented in the medical records, which can cause underreporting, and some cases were analyzed through interviews. Therefore, recall bias was possibly present. Third, we obtained samples at the estimated times at which the apixaban plasma levels were at their peak; at 2–4 h, there were no data available for peak levels for each patient. Finally, our study compared the apixaban plasma levels with the expected range presented in clinical practice guidelines [11]. The exact therapeutic range of the apixaban plasma levels was not defined, while intra-individual variability in the apixaban plasma levels was already reported in another study [9,13].

## 5. Conclusions

In summary, almost every Thai patient with NVAF was in the unexpected range of peak and trough apixaban plasma levels during all dosage regimens. Our study is the first to investigate the apixaban plasma levels in a developing Asian country and observe that bleeding events are associated with the trough of apixaban plasma level and bleeding history. While stroke events could occur, the apixaban plasma levels were unexpected in the previously described ranges, and it was noticeable that all patients in the stroke group had a history of stroke.

## Figures and Tables

**Figure 1 biomedicines-10-02001-f001:**
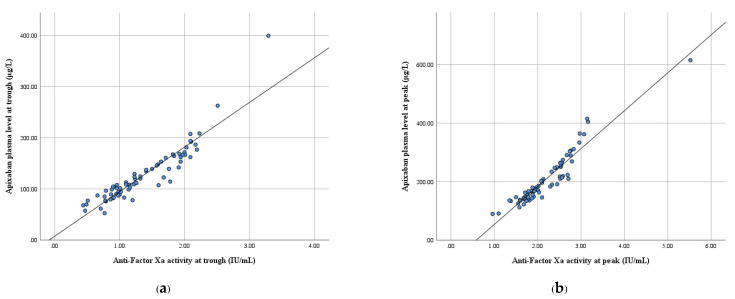
Scatterplot between the apixaban plasma level and anti-FXa activity. (**a**) At the trough and (**b**) at the peak.

**Table 1 biomedicines-10-02001-t001:** Baseline characteristics for all patients.

Characteristics	All Patients (*n* = 71)
Age (years)	75.0 ± 10.5
Age ≥ 80 years	26 (36.6%)
Male sex	49 (69.0%)
Body weight (kg)	67.3 ± 13.5
BMI (kg/m^2^)	24.6 ± 3.9
SCr (mg/dL)	1.1 (0.9–1.4)
CrCl (mL/min)	54.1 ± 22.0
CHA_2_DS_2_-VASc score	4 (3–5)
HAS-BLED score	2 (1–3)
Bleeding history	29 (40.3%)
Comorbidities	
Hypertension	62 (83.3%)
Dyslipidemia	50 (70.4%)
Anemia	30 (42.3%)
Diabetic mellitus	28 (39.4%)
Stroke	27 (38.0%)
Chronic kidney disease	26 (36.6%)
Heart failure	24 (33.8%)
Interacting drugs	
Amiodarone	11 (15.5%)
Antiplatelets	7 (9.9%)
Dronedarone	3 (4.2%)
Propafenone	2 (2.8%)
Ranolazine	2 (2.8%)
Valproic acid	1 (1.4%)

Abbreviations: BMI, body mass index; kg, kilogram; SCr, serum creatinine; CrCl, creatinine clearance by the Cockcroft–Gault equation; CHA_2_DS_2_-VASC, congestive heart failure, hypertension, age (≥75 years), diabetes, stroke/transient ischemic attack, vascular disease (peripheral arterial disease, previous myocardial infarction, or aortic atheroma), age (≥65 years), and sex category; HAS-BLED, uncontrolled blood pressure, abnormal renal or liver function, stroke history, bleeding tendency or predisposition, labile international normalized ratio, age (≥65 years), medication (aspirin, clopidogrel, or non-steroidal anti-inflammatory drugs), and alcohol use.

**Table 2 biomedicines-10-02001-t002:** The trough and peak values of the apixaban plasma levels and anti-FXa activities at steady state.

Parameters	* Dose of Apixaban		Overall (*n* = 71)	*p*-Value
	Reduced Dose (*n* = 25)	Standard Dose (*n* = 46)		
C_trough_ (μg/L)	120.21	110.81	112.79	0.791
	(76.01–168.26)	(63.08–249.01)	(68.69–207.82)	
C_peak_ (μg/L)	170.89	205.44	185.62	0.159
	(127.42–269.69)	(125.18–412.31)	(124.06–384.34)	
Xa_trough_ (IU/mL)	1.23	1.23	1.23	0.605
	(0.54–2.07)	(0.73–2.44)	(0.59–2.21)	
Xa_peak_ (IU/mL)	1.94	2.21	2.10	0.093
	(1.36–2.78)	(1.61–3.16)	(1.44–3.11)	

Abbreviations: C_peak_, apixaban peak plasma level; C_trough_, apixaban trough plasma level; Xa_peak_, anti-Factor Xa activity at peak level; Xa_trough_, anti-Factor Xa activity at trough level. Data are presented as the median (5th–95th percentile range). * The standard dose of apixaban was defined as orally ingesting 5 mg twice daily. If patients fell into at least two criteria, such as age ≥ 80 years, body weight ≤ 60 kg, or SCr ≥ 1.5 mg/dL, they received a reduced dose of apixaban (2.5 mg), orally, twice daily [1].

**Table 3 biomedicines-10-02001-t003:** The trough and peak values of the apixaban plasma levels classified by stroke events.

Parameter	Stroke (*n* = 8)	Non-Stroke (*n* = 63)	*p*-Value
C_trough_ (μg/L)	146.24	111.48	0.604
	[92.24, 163.22]	[89.61, 156.92]	
C_peak_ (μg/L)	200.52	183.05	0.834
	[161.61, 259.50]	[146.00, 249.74]	

Abbreviations: C_peak_, apixaban peak plasma level; C_trough_, apixaban trough plasma level. Data are presented as the median [IQR].

**Table 4 biomedicines-10-02001-t004:** The trough and peak values of the apixaban plasma levels classified by bleeding events.

Parameters	Bleeding Events				*p*-Value	
	Overall Bleeding (*n* = 14)	Major Bleeding (*n* = 6)	Minor Bleeding (*n* = 8)	Non- Bleeding (*n* = 57)		
C_trough_ (μg/L)	139.15 [109.49, 163.16]	123.95 [93.81–139.19]	149.46 [117.42, 172.89]	108.14 [87.23, 160.59]	0.126 *	0.148 ^†^
C_peak_ (μg/L)	209.32 [145.23, 285.45]	178.66 [137.49–209.76]	269.30 [157.83–317.69]	183.05 [147.08, 247.03]	0.470 *	0.330 ^†^

Abbreviations: C_peak_, apixaban peak plasma level; C_trough_, apixaban trough plasma level. Data are presented as the median [IQR]. * Comparison between overall and non-bleeding groups. ^†^ Comparison between major, minor, and non-bleeding groups.

**Table 5 biomedicines-10-02001-t005:** Multivariate analysis of the apixaban plasma levels related to bleeding events.

Clinical Factors	At Trough		At Peak	
	OR (95% CI)	*p*-Value	OR (95% CI)	*p*-Value
Bleeding history	17.62 (3.54–176.64)	0.002	17.26 (3.57–157.38)	0.002
Apixaban plasma level	1.01 (1.00–1.03)	0.038	1.01 (1.00–1.02)	0.050

Abbreviations: OR, odds ratio; 95% CI, 95% confidence interval.

## Data Availability

Not applicable.

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
