# Peer review of "Do Apixaban Plasma Levels Relate to Bleeding? The Clinical Outcomes and Predictive Factors for Bleeding in Patients with Non-Valvular Atrial Fibrillation"

_biomedicines, 2022, doi:10.3390/biomedicines10082001_

Round 1
Reviewer 1 Report
This manuscript addresses the association between blood levels of Apixaban and complicated bleeding in patients with non-valvular atrial fibrillation. One very interesting finding was that trough blood levels tended to be lower in the haemorrhage and non-bleeding groups, but did not differ significantly. However, multivariate analysis concluded that a history of bleeding and trough blood levels were associated with bleeding events. This is a very interesting finding.
We would like to comment on some points that are unclear about this paper.
a) The ESC guidelines generally use the HAS-BLED score, but this multivariate analysis does not include the factors hypertension, renal dysfunction, liver dysfunction, cerebral embolisation, haemorrhage, older age, drugs and alcohol, which factors were incorporated in the results? The results of the univariate analysis would also be interesting and should be presented.
b) It would be very useful if this could be done using trough blood levels as a predictor of bleeding events. Are there any factors that could be biases associated with this Apixaban trough blood level? Do you think that renal dysfunction, for example, could have an impact?
c) This study was conducted in Bangkok, Thailand - are all patients of Asian ethnicity? What is the breakdown of this Asian population?
Author Response
Dear Reviewer,
Thank you for your valuable comments. We have already corrected the manuscript as your suggestion.
Comments and Suggestions for Authors
This manuscript addresses the association between blood levels of Apixaban and complicated bleeding in patients with non-valvular atrial fibrillation. One very interesting finding was that trough blood levels tended to be lower in the haemorrhage and non-bleeding groups, but did not differ significantly. However, multivariate analysis concluded that a history of bleeding and trough blood levels were associated with bleeding events. This is a very interesting finding.
We would like to comment on some points that are unclear about this paper.
Comment # a) The ESC guidelines generally use the HAS-BLED score, but this multivariate analysis does not include the factors hypertension, renal dysfunction, liver dysfunction, cerebral embolisation, haemorrhage, older age, drugs and alcohol, which factors were incorporated in the results? The results of the univariate analysis would also be interesting and should be presented.
Response: Thank you for your valuable comments. We have already corrected the manuscript as your suggestion. Therefore, we provided the sentence “Based on the univariate analysis, there were eight factors that demonstrated po-tential effects on bleeding events, including hypertension, renal dysfunction, stroke, older age, drugs, alcohol, and apixaban plasma level. The results of the multivariate analysis suggest that the bleeding history is only one factor that is significant for in-creasing bleeding risk (OR 13.00; 95% CI 2.63-64.24; p-value = 0.002), in section 3.4. Relationship between Clinical Factors and Clinical Events, on page 7.
Comment # b) It would be very useful if this could be done using trough blood levels as a predictor of bleeding events. Are there any factors that could be biases associated with this Apixaban trough blood level? Do you think that renal dysfunction, for example, could have an impact?
Response:
Thank you for your valuable comments. The authors think that there any factors that could be biases associated with apixaban trough level. Therefore, we have already corrected the manuscript as your suggestion as following:
We added the results that “For the correlation between apixaban plasma levels and clinical factors, only CrCl (correlation coefficient, ρ = -0.27; p-value < 0.024) and the concomitant drugs with CYP3A4 or P-gp inhibitors (rpb=0.32; p-value < 0.006) were predictors for apixaban through plasma level. Whereas, there was no the correlation between apixaban peak plasma levels and any factors” in Results section (3.2 Steady-State Apixaban Plasma Levels and Anti-FXa Activities, on page 5. Therefore, we added this sentence “The correlation between apixaban plasma levels and clinical factors were analyzed using Spearman’s rank (ρ) and point biserial correlation (rbp).” in section 2.4 Statistical Analysis, on page 3.
In addition, we think that renal dysfunction could be effect on apixaban level. Therefore, we have already corrected the manuscript as your suggestion as following: this sentence “In addition, a previous study reported that apixaban concomitant with CYP3A or P-gp inhibitor drugs was not related to higher plasma levels in patients with AF [23]. However, our study presented a positive correlation between the concomitant CYP3A or P-gp inhibitor drugs and Ctrough. Amiodarone and dronedarone are moderate CYP3A4 and P-gp inhibitors that may increase apixaban plasma levels [24,25]. These results con-form to the results obtained from a previous pharmacokinetic study that suggested that strong-to-moderate CYP3A and P-gp inhibitors affect the AUC and apixaban plas-ma levels [26]. Furthermore, approximately 27% of apixaban was excreted unchanged by the kidneys. This pharmacokinetic finding illustrates that the impact of renal impair-ment on apixaban exposure is modest [27]. A regression analysis showed that AUC0–∞ was inversely related to apixaban clearance in renal impairment compared to healthy volunteers having normal renal functions [7,28]. Testa et al. provided data from a re-al-world study that determined that CrCl has a poor correlation with apixaban plasma levels [9]. Similarly to the present study, they observed a weak inverse correlation be-tween CrCl and trough of apixaban plasma levels.” was added in discussion section, on page 9.
Comment # c) This study was conducted in Bangkok, Thailand - are all patients of Asian ethnicity? What is the breakdown of this Asian population?
Response:
We apologized for the unclear content and thank you for your valuable comments. All patients in our study are Thai patients. We mentioned “Asian population” because some studies found that Asians have higher exposure to apixaban than non-Asians, which could increase the bleeding risk in Asian population because of genetic polymorphisms. In addition, the results of this study may be useful for other Asian countries.
Sincerely,
Pornwalai Boonmuang

Reviewer 2 Report
This is a clinical study, which aimed to investigate the clinical outcomes among NVAF patients in real-life clinical practice receiving apixaban and the risk factors associated bleeding events especially factor of apixaban plasma level. The authors concluded that bleeding events were associated with the trough of apixaban plasma level and bleeding history.
This is clinically important, and this reviewer has some comments as described below.
Major comments:
1. The authors defined the peak time of apixaban plasma levels as 2-4 hours post dose administration; however, there were no data that the time was really peak in each patient. The peak time is variable in patients. Therefore, the authors should mention this issue in the Limitation section.
2. In the Methods section, the authors should briefly describe what were major and minor bleeding.
3. Chromogenic anti-FXa assay. Did the authors perform this assay in both peak and trough measurement of apixaban plasma levels? The authors should clearly describe if they performed simultaneously examined the apixaban plasma levels and chromogenic anti-FXa assay.
4. Results section 3.3. Was stroke only brain infarction? Does stroke involve brain bleeding?
5. Table 3. Is it possible to analyze the bleeding events separately in major (n=6) and minor events (n=8)?
6. What were predictive factors in stroke?
Minor comment:
7. Page 6. “3.3. Correlation between Apixaban Plasma Level and LMWH-calibrated Anti-FXa activity” supposed to be “3.5.”
Author Response
Dear Reviewer:
Thank you for your valuable comments. We have already to corrected the MS as your suggestion. (Please see the attachment)
Major comments:
Comment # 1: The authors defined the peak time of apixaban plasma levels as 2-4 hours post dose administration; however, there were no data that the time was really peak in each patient. The peak time is variable in patients. Therefore, the authors should mention this issue in the Limitation section.
Response:
We totally agree with your suggestion. Therefore, we provided the sentence “Third, we obtained sampled at the estimated in which apixaban plasma levels were at their peak; at 2–4 hours, there were no data available for peak levels for each patient.” in Limitation section, on page 9.
Comment # 2 In the Methods section, the authors should briefly describe what were major and minor bleeding.
Response:
Thank you for your suggestion. We have already corrected the manuscript as your suggestion as following: we added the sentence “(major bleeding was defined in nonsurgical patients as fatal bleeding, symptomatic bleeding in a critical area or organ, and/or bleeding causing a decline in hemoglobin levels of 2 g/dL or more, or leading to the transfusion of two or more units of whole blood or red cells; minor bleeding was defined as “non-major bleeding”, on page 2.
Comment # 3 Chromogenic anti-FXa assay. Did the authors perform this assay in both peak and trough measurement of apixaban plasma levels? The authors should clearly describe if they performed simultaneously examined the apixaban plasma levels and chromogenic anti-FXa assay.
Response:
Thank you for your valuable comments. This study performed Chromogenic anti-FXa assay in both peak and trough measurement of apixaban plasma levels. We have already corrected the manuscript as your suggestion as following:
We are very appreciated to correct this comment because it could be more obvious for the reader. Therefore, we changed the sentence “At a steady state, apixaban plasma level and anti-FXa activity were measured at peak and trough concentrations. Blood samples were collected at the peak and trough measurements. This study defined trough time as the time immediately before apixaban intake and the peak time as 2-4 hours post dose administration of apixaban. Blood samples were collected into two 3.2% citrated tubes for trough and peak. Immediately, the samples were centrifuged for 15 minutes at 2500-3000 × g [15,16]. Thereupon, apixaban plasma level and LMWH-calibrated anti-FXa activity were measured using a chromo-genic anti-FXa assay with the BiophenTM heparin LRT kit (Hyphen BioMed, Neuville-sur-Oise, France) and analyzed by a Sysmex CS 2500 System (Siemens Health Care, Milan, Italy), which was calibrated with commercial apixaban (sensitivity range 0-600 μg/L) and LMWH (sensitivity range 0-1.75 IU/mL). Chromogenic anti-FXa assay was based on p-nitroaniline release from a specific chromogenic factor Xa substrate. Consequently, the optical density generated per minute (OD/min) was inversely proportional to the amount of direct factor Xa inhibitor in the sample. Therefore, chromogenic anti-Factor Xa assay was the most suitable for the rapid quantitation of apixaban [16].” to “At a steady state, apixaban plasma level and anti-FXa activity were measured at peak and trough concentrations by Chromogenic anti-FXa assay. Blood samples were collected at the peak and trough measurements. This study defined trough time as the time immediately before apixaban intake and the peak time as 2-4 hours post dose administration of apixaban. Blood samples were collected into two 3.2% citrated tubes for trough and peak. Immediately, the samples were centrifuged for 15 minutes at 2500-3000 × g [15,16]. Subsequently, the samples were measured for their anti-FXa activity using a chromo-genic anti-FXa assay. Anti-FXa activity was determined using the BiophenTM heparin LRT kit (Hyphen BioMed, Neuville-sur-Oise, France) and analyzed by a Sysmex CS 2500 System (Siemens Health Care, Milan, Italy). The anti-Xa activity results obtained from the chromogenic assay were converted into commercial apixaban (sensitivity range: 0-600 μg/L) and LMWH units (sensitivity range: 0-1.75 IU/mL) [16]., on page 3.
Comment # 4 Results section 3.3. Was stroke only brain infarction? Does stroke involve brain bleeding?
Response:
We apologized for this that it was unclear. Stroke outcomes in this study defined as ischemic stroke (brain infarction) and hemorrhagic stroke. Therefore, we changed the sentence “The secondary outcomes were analyzed as follows: (1) stroke and bleeding events: stroke defined as ischemic and hemorrhagic strokes, including a definition of bleeding that was based on the criteria of the International Society on Thrombosis and Haemostasis (ISTH)” in section 2.2 Primary and Secondary outcomes, on page 2.
Comment # 5 Table 3. Is it possible to analyze the bleeding events separately in major (n=6) and minor events (n=8)?
Response:
Thank you for your valuable comments. We have already corrected the manuscript as your suggestion. We added the new results in the sentence “The median values of Ctrough in the minor, major, and non-bleeding groups were 149.46 (IQR 117.42, 172.89) μg/L, 123.95 (IQR 93.81-139.19) μg/L, and 108.14 (IQR 87.23-160.59) μg/L, respectively, with no significant differences (p-value=0.148). The median of Cpeak in the minor, major, and non-bleeding groups were 269.30 (IQR 157.83, 317.69) μg/L, 178.66 (IQR 137.49-209.76) μg/L, and 183.05 (IQR 147.08-247.03) μg/L, respectively, with no significant differences (p-value=0.330). The trough and peak values of apixaban plasma lev-els classified by bleeding events are also presented in Table 4.” in Results section, on page 6. In addition, we provided the new results in Table 4.
Comment # 6 What were predictive factors in stroke?
Response:
Thank you for your suggestion. This study analyzed the predictive factors in stroke based on CHA2DS2VASC score as the stroke risk assessment tool. Therefore, the authors added this sentence “Additionally, in the univariate analysis, there were eight factors that demonstrated potential effects on stroke events, including hypertension, heart failure, previous stroke, older age, vascular disease, diabetes mellitus, CHA2DS2VASC score, and apixaban plasma level. The results reveal that the CHA2DS2VASC score (OR 1.90; 95% CI 1.15-3.14; p-value = 0.013) is significantly related to stroke events. Unfortunately, the multivariate analysis results do not demonstrate any factors that presented significant differences for stroke events.” in Results section (3.4 Relationship between Clinical Factors and Bleeding Events), on page 7.
Minor comment:
Comment # 7 Page 6. “3.3. Correlation between Apixaban Plasma Level and LMWH-calibrated Anti-FXa activity” supposed to be “3.5.”
Response:
We apologized for this and thank you for your suggestion. The phrase "3.3. Correlation between Apixaban Plasma Level and LMWH-calibrated Anti-FXa activity" was changed into “3.5. Correlation between Apixaban Plasma Level and LMWH-calibrated Anti-FXa activity” on page 7.

Round 2
Reviewer 1 Report
Although there are no comments on the reviewers' questions, I believe that the minimum criteria for acceptance have been met.
Reviewer 2 Report
This reviewer has no further comment.